# The Effect of Cooling Layer Thickness and Coolant Velocity on Crystal Thermodynamic Properties in a Laser Amplifier

**DOI:** 10.3390/mi14020299

**Published:** 2023-01-23

**Authors:** Shuzhen Nie, Tianzhuo Zhao, Xiaolong Liu, Pubo Qu, Yuchuan Yang, Yuheng Wang

**Affiliations:** 1Aerospace Information Research Institute, Chinese Academy of Sciences, Beijing 100094, China; 2School of Optoelectronics, University of Chinese Academy of Sciences, Beijing 100049, China; 3School of Artificial Intelligence, Henan University, Zhengzhou 450046, China

**Keywords:** LD side-pumped laser amplifier, thermal analysis, cooling layer thickness, coolant velocity, convective heat transfer coefficient, inlet pressure

## Abstract

Laser diode pumped solid-state lasers (DPSSLs) have been widely used in many fields, and their thermal effects have attracted more and more attention. The laser diode (LD) side-pumped amplifier, as a key component of DPSSLs, is necessary for effective heat dissipation. In this paper, instead of the common thermal analysis based only on a crystal rod model, a fluid–structure interaction model including a glass tube, cooling channel, coolant and crystal rod is established in numerical simulation using ANSYS FLUENT for the configuration of an LD array side-pumped laser amplifier. The relationships between cooling layer thickness, coolant velocity and maximum temperature, maximum equivalent stress, inlet pressure and the convective heat transfer coefficient are analyzed. The results show that the maximum temperature (or maximum equivalent stress) decreases with the increase in the coolant velocity; at low velocity, a larger cooling layer thickness with more coolant is not conductive enough for improved heat dissipation of the crystal rod; at high velocity, when the cooling layer thickness is above or below 1.5 mm, the influence of the cooling layer thickness on the maximum temperature can be ignored; and the effect of the cooling layer thickness on the maximum equivalent stress at high velocity is not very significant. The comprehensive influence of various factors should be fully considered in the design process, and this study provides an important reference for the design and optimization of a laser amplifier and DPSSL system.

## 1. Introduction

As one of the development frontiers of laser technology, by means of spatial filtering, structure optimization, beam modulation, etc., the laser diode pumped solid-state laser (DPSSL) has gradually gained the advantages of high conversion efficiency, large output energy, good beam quality, compact structure, repeatable frequency operation, etc. [1]. Therefore, it has important applications in precision detection, advanced manufacturing, scientific instruments, leading-edge science and other fields [2,3,4]. The thermal problem is an important bottleneck that hinders DPSSL development regarding higher average power and better beam quality, because when the gain medium is pumped by a high-power laser diode (LD), a large amount of useless heat will be generated [5,6]. The existence of useless heat will lead to thermal lens, thermal stress, depolarization, birefringence and other effects, thus limiting DPSSLs’ ability to achieve higher output power or better beam quality, and even damaging the gain medium, which seriously affects the improvement of the DPSSL system [7,8].

DPSSLs mainly adopt the technical scheme of “oscillation and amplification” to achieve high output power, which is called the master oscillator power amplifier (MOPA) system [9]. Oscillator technology is relatively mature; due to its low average power, the heat dissipation of the oscillator is sufficient. Therefore, effective heat dissipation of the amplifier is the key to achieving higher DPSSL output performance. At present, amplifiers in high-power DPSSL systems commonly use an LD side-pumped configuration, and the gain medium is a crystal rod or slab (rectangular prism) [10,11,12]. The configuration with multiple LD arrays side-pumped and using a crystal rod is relatively simple and reliable, providing high pumped LD energy, large aperture scalability and smooth pumped power distribution along the crystal length, so high average output power or pulse energy can be achieved [13,14]. However, the spatial and geometric structures of the side-pumped LD array and gain medium seriously affect the quality of the output beam [15]. Therefore, more attention must be paid to the configuration design process, especially the impact on thermal effects.

Previous work on analyses of LD side-pumped laser rods has shown that overcoming this design challenge necessitates knowledge of the temperature distribution within the gain medium [16,17,18,19]. Dai et al. [20] used a triangular LD side-pumped structure and established a numerical temperature model of the crystal rod using finite element analysis. The asymmetry of temperature distribution in the x and y directions was obtained mainly because of the triangle pumped structure. YE et al. [21] adopted a five-LD-array side-pumped structure and analyzed the influence of crystal size and LD distribution on the resulting crystal rod temperature using ray tracing and the finite element method. The thermal effect on the crystal still affected laser performance. Zhang et al. [22] compared three-, five- and nine-LD-array side-pumped structures to achieve higher pumping homogeneity. It was found that more uniform gain test results can be obtained by using the nine-LD-array structure. The ring-LDA side-pumped structure was studied by Zhang et al. [23]. They analyzed and compared the transient temperature distribution in the crystal rod under different pump periods, pump frequencies and pump widths. Wang et al. [24] discussed three different side-pumped geometries: the segmented circular LD array side-pumped configuration, the annular liquid-cooling structure and the compensated semicircular LD array side-pumped arrangement. The temperature distribution of the laser rod was analyzed using a numerical analysis method, which indicated that the side-pumped configuration of the segmented circular LD array provided higher beam quality. In this research work, it was found that using more LD array side-pumped configurations leads to better output effects, and the temperature distribution is always based on the modeling of the crystal rod using finite element calculation tools. The effective heat dissipation of the coolant is achieved by adding the convective heat transfer coefficient as the boundary condition on the surface of the crystal rod during the modeling process. However, in the side-pumped configuration using more LD arrays, the heat dissipation effect of the coolant on the crystal rod is mainly determined by the coolant parameters and the cooling channel structure. Inlet pressure, fluid velocity, cooling temperature and other parameters related to the coolant in the cooling channel also need to be considered. If these factors are included in the numerical simulation of the thermal analysis, more accurate results will be obtained. However, little research has been carried out on this topic.

In this paper, a fluid–structure interaction model including a glass tube, cooling channel, coolant and crystal rod is established in ANSYS FLUENT for the configuration of an LD array side-pumped laser amplifier. The effects of cooling layer thickness and coolant velocity on the maximum temperature and maximum equivalent stress of the crystal rod are systematically discussed. Inlet pressures under different cooling layer thicknesses and coolant velocities are also obtained, and the convective heat transfer coefficient calculated by the model is compared with the result of the equation solution. The analysis results provide important guidance for the design and optimization of cooling parameters of laser amplifiers and DPSSL systems.

## 2. Theoretical Model and Parameters

### 2.1. Theory

The mechanism by which the coolant cools the crystal rod involves coolant flow and heat transfer, coupled heat transfer at the interface and heat conduction of the solid structure. Based on computational fluid dynamics and numerical heat transfer, the governing equations of fluid and heat transfer, including the mass equation, momentum equation and energy equation of fluid, as well as the heat conduction equation of solids, can be expressed in the following general form [25]:(1)$$\frac{\partial \rho_{f}}{\partial t}+\nabla \cdot \left(\rho_{f}u\right)=0$$
(2)$$\frac{\partial \left(\rho_{f}u_{i}\right)}{\partial t}+\nabla \cdot \left(\rho_{f}u_{i}u\right)=-\frac{\partial P}{\partial x_{i}}+\nabla \cdot \left(\mu \nabla u_{i}\right)$$
(3)$$\frac{\partial \left(\rho_{f}T\right)}{\partial t}+\nabla \cdot \left(\rho_{f}Tu\right)=\nabla \cdot \left(\frac{k_{f}}{C_{f}}\nabla T\right)+Q_{f}$$
where $\rho_{f}$ is the fluid density, u is the velocity vector of the fluid, $u_{i}$ (*i* = 1, 2, 3) is the velocity component in the x, y and z directions, *P* is the pressure of the fluid, and $x_{i}$(*i* = 1, 2, 3) represents x, y, and z, 
respectively. *μ* is the viscosity coefficient of the coolant. $k_{f}$ is the thermal 
conductivity of the fluid and $C_{f}$ is the specific heat of the fluid, *T* is the temperature, and $Q_{f}$ is the heat source in the fluid. If the velocity in Equation (3) is zero, the heat conduction equation in the solid can be expressed as:(4)$$\frac{\partial \left(\rho_{s}T\right)}{\partial t}-\nabla \cdot \left(\frac{k_{s}}{C_{s}}\nabla T\right)=Q_{s}$$
where $\rho_{s}$, $k_{s}$ and $C_{s}$ are the density, thermal conductivity and specific heat, respectively, of the solid. $Q_{s}$ is the heat source in the solid. The thermal coupling conditions on the fluid–solid coupling boundary are
(5)$$\left\{\begin{array}{l} T_{s}=T_{f};\\ k_{s}\left(\nabla T_{s}\right)\cdot n_{s}=k_{f}\left(\nabla T_{f}\right)\cdot n_{f} \end{array}\right.$$
where $n_{s}$ and $n_{f}$ are the outer normal vectors of the solid–fluid coupling boundary. $T_{s}$ and $T_{f}$ are the temperatures of the solid and fluid on the fluid–solid coupling boundary. Based on the above formulas, the distribution of the coolant flow field and the temperature distribution of the whole system can be obtained simultaneously. For the relatively complex coupling of multiple physical fields involved in computational fluid dynamics and numerical heat transfer, ANSYS FLUENT is used here for simulation analysis.

The pumped LD laser is used as the internal heat source in the loading regions of the crystal rod, and the fluid velocity at the inlet is used as the input condition. The crystal surfaces with coolant flowing through conform to the interface setting of the boundary conditions of fluid–structure interaction. Coolant-free surfaces adopt the boundary conditions of natural convection heat exchange with air. The model’s initial temperature is usually room temperature.

### 2.2. Modeling

The fluid–structure interaction model with the glass tube, cooling channel, coolant and crystal rod is shown in Figure 1. The crystal rod and the glass tube are placed coaxially. The length of the crystal and the glass tube is 100 mm and 63 mm. The cooling channel is filled with coolant, which flows in from one end of the glass tube and out from the other end. Both ends are connected to the external structure modules to realize the circulation of coolant. Therefore, only the part of the rod inside the glass tube is cooled by the coolant, while the other part is placed in the air to install the fixed structure of the crystal rod. The convective heat transfer coefficient of air should be 1–10 W/m^2^/K [26], and here we use 6.5 W/m^2^/K according to the empirical value.

There are four groups of LD arrays with a wavelength of 802 ± 1 nm, and an average power of 1.25 W is used to irradiate each group on the crystal rod, which has a diameter of 5 mm. The loaded regions and structure with a group of LD arrays are shown in Figure 2. The input laser power is considered the internal heat source of each region. When the total temperature fluctuation is not too large, it is assumed that the material parameters of the crystal, glass tube and coolant are constant, the pumped laser distribution is uniform in the direction of the crystal optical axis, the output power and spatial distribution of the laser diode array are the same in each group, the coolant and glass tube are transparent with no absorption in the pumped laser wavelength range, and no other thermal effects are induced by the side-pumped laser except the crystal rod. The model parameters are shown in Table 1. The initial and ambient temperature is 293.15 K. The input coolant temperature is 293.15 K. Gravity is considered.

## 3. Results and Discussions

### 3.1. Simulation

The model is meshed using the numerical simulation tool (FLUENT MESHING) and the element size of the crystal rod and glass tube is 0.5 mm, as shown in Figure 3. Taking into account the calculation accuracy and period, the element size of the cooling layer is selected based on the actual size of the cooling channel. The wall thickness of the glass tube is 1 mm. The cooling layer thickness (expressed as df) in the cooling channel is 300 μm, 500 μm, 800 μm, 1 mm, 1.5 mm, 2 mm and 3 mm. The coolant velocity (expressed as V) is 1.5 m/s, 1.6 m/s, 1.7 m/s, 2 m/s, 2.5 m/s, 3 m/s, 3.5 m/s, 4 m/s and 4.5 m/s. After modeling, grid generation and boundary conditions are applied, the solution process can be carried out by appropriately selecting k-epsilon solvers and wall functions. When the coolant velocity and the cooling layer thickness are different, the heat dissipation effect will also change and the relationship between them is revealed.

### 3.2. Temperature Results

In the case of different coolant velocities and cooling layer thicknesses, the temperature distribution on the crystal rod is similar. Three regions are selected to investigate temperature results at different locations of the crystal rod, as shown in Figure 4a. Region 1 is irradiated with the laser and the temperature field in the cross-section (V = 4.5 m/s, df = 1 mm) is shown in Figure 4b. The maximum temperature (expressed as Tmax), 311.15 K, is located in the center of the rod. Region 2 is cooled by the coolant flow without laser irradiation, and Region 3 is placed in the air. The cross-sections of Region 2 and Region 3 have similar temperature distributions, as shown in Figure 4b. Maximum temperatures of 302.22 K and 293.16 K are in the center. It can be found that the temperature difference between the center and the edge of the crystal rod in the laser-loaded region is larger than that in other regions. Temperature profiles along the axis in Region 2 and Region 3 are shown in Figure 4c,d. It can be observed that the temperature differences are very small in the boundary of the coolant and air as boundary conditions. The temperature decreases gradually when it is far away from the laser-loaded region. The effect of cooling layer thickness on the maximum temperature at different coolant velocities is shown in Figure 5.

Figure 5 shows that with the increase in the coolant velocity, the maximum temperature under different cooling layer thicknesses is characterized by a continuous decrease. Therefore, high coolant velocity is helpful for effective heat dissipation of the crystal rod. At the same lower coolant velocity (V < 2.5 m/s), the larger the layer thickness, the higher the maximum temperature of the crystal rod. It can be noted that under the condition of constant low velocity, a larger cooling layer thickness with more coolant is not conductive enough to improve heat dissipation for the crystal rod. As V = 2.5 m/s, the maximum temperature is similar when the cooling layer thickness is 1 mm and 800 μm. When V > 2.5 m/s, some changing trends can be observed; the maximum temperature is closer when the cooling layer thickness is less than 1.5 mm or more than 1.5 mm. This means that in the case of high velocity, with df = 1.5 mm as the dividing line, when the cooling layer thickness is above or below 1.5 mm, the influence of the cooling layer thickness on the maximum temperature can be ignored. Therefore, at high coolant velocity, it is not necessary to choose a particularly small or large cooling layer thickness in order to obtain better heat dissipation performance, and their differences are less than 0.5 K.

### 3.3. Stress Results

According to the temperature results, the thermal stress can be calculated by adding the constraint conditions to the crystal part in the air. This part is connected to the fixed structure. Maximum equivalent stress is simulated as the coolant velocity increases for various cooling layer thicknesses, as shown in Figure 6. It can be found that under different cooling layer thicknesses, the maximum equivalent stress decreases with the increase in the coolant velocity. The variation trend of maximum equivalent stress with coolant velocity coincides with that of maximum temperature. At the same coolant velocity, the maximum equivalent stress increases with the increase in the coolant layer thickness. When V > 2.5 m/s, the maximum equivalent stress is similar to the cooling layer thicknesses of 1.5 mm and 2 mm. It can also be observed that as the coolant velocity increases, the maximum equivalent stress differences under different cooling layer thicknesses become smaller. Therefore, at high coolant velocity, the effect of the cooling layer thickness on the maximum equivalent stress is not very significant, with differences of less than 1 MPa.

### 3.4. Inlet Pressure

The inlet pressure of the configuration must be considered because high inlet pressure will lead to sealing and assembly difficulties. The calculation of inlet pressure is based on Bernoulli’s Equation (6):(6)$$P+\frac{1}{2}\rho v^{2}+\rho gz=P_{t}$$
where *P* is the static pressure, *v* is the fluid velocity, *ρ* is the fluid density, *g* is the acceleration of gravity, and *z* is the height above an arbitrary reference level. *P_t_* is total pressure. In the fluid–structure interaction model, the inlet pressure is considered total inlet pressure, which can be obtained in numerical simulation.

The inlet pressure results for varying coolant velocity and cooling layer thickness values are presented in Figure 7. Figure 7a shows that under different cooling layer thicknesses, the inlet pressure increases with the increase in the coolant velocity. For values of constant smaller cooling layer thickness, the inlet pressure is much more sensitive to the effect of coolant velocity. At the same coolant velocity, the inlet pressure decreases as the coolant layer thickness increases. As the coolant velocity increases, the differences in inlet pressure under different cooling layer thicknesses become larger. This means that although the smaller cooling layer thickness will result in a lower maximum temperature and thermal stress, it will significantly increase the inlet pressure, thus increasing the risk of coolant leakage. Therefore, in the design process of an amplifier configuration, the above factors need to be considered comprehensively.

### 3.5. Convective Heat Transfer Coefficients

The convective heat transfer coefficient is an important parameter in the thermal management of DPSSL design. For LD side-pumped laser amplifiers, the convective heat transfer coefficient on the outer surface of the crystal rod represents the cooling capacity as the coolant flows through. This parameter is related to the properties of the coolant and the structure of the cooling channel. Although its value is very important, it is relatively difficult to determine its value [27], and there is an inconsistency between different mathematical models. For this configuration, according to the Reynolds number of the coolant in the cooling channel, Equation (7) of the surface convective heat transfer coefficient is selected from [28]. When the Reynolds number is greater than 2100 and less than 12,000, we replace the inner tube diameter with the equivalent annular diameter to approximate the solution in Equation (7), and precise equations could be further studied.
(7)$$\left\{\begin{array}{ll} h=1.02\frac{k}{D_{2}-D_{1}}N_{Re}{}^{0.45}N_{Pr}{}^{0.5}N_{Gr}{}^{0.05}{\left(\frac{D_{2}-D_{1}}{L}\right)}^{0.4}{\left(\frac{D_{2}}{D_{1}}\right)}^{0.8} & 900<N_{Re}<2000\\ h=0.023\frac{k}{D_{2}-D_{1}}N_{Re}{}^{0.8}N_{Pr}{}^{0.4} & 2100<N_{Re}<12000\\ N_{Re}=(D_{2}-D_{1})V/\mu ;N_{Pr}=C\mu /k;N_{Gr}={\left(D_{2}-D_{1}\right)}^{3}\rho^{2}gr\Delta T/\mu^{2} \end{array}\right.$$
where *k* is the thermal conductivity of the coolant, *D*_2_ is the inner diameter of the glass tube, *D*_1_ is the diameter of the crystal rod, and *L* is the length of the cooling channel. *N_Re_* is Reynolds number, *N_Pr_* is the Prandtl number, and *N_Gr_* is the Grashof number. *V*, μ, *C*, ρ, and *r* are the velocity, viscosity, specific heat, density and thermal expansion coefficient of the coolant, respectively. *g* is the gravitational constant and
ΔT is the temperature difference between the rod surface and the coolant.

The solution results of the convective heat transfer coefficient according to Equation (7) are shown in Figure 8a. The results indicate that under different cooling layer thicknesses, the convective heat transfer coefficient increases with the increase in coolant velocity. Because of the different Reynolds number ranges, the curves are correspondingly different. For df = 300 μm, the convective heat transfer coefficient has a great increase from V = 2.5 m/s to V = 3 m/s. For df = 1.5 mm, the convective heat transfer coefficient has a slight rise from V = 3 m/s to V = 3.5 m/s. For df > 1 mm, convective heat transfer coefficients become similar at V > 3 m/s. Comparative results of simulation and equation solutions are shown in Figure 8b–h. It can be found that when df < 1.5 mm, the equation solution commonly demonstrates higher values than the simulation result, while when df > 1.5 mm, it is the opposite. The deviation between them is mostly within 10%, so in direct solid thermodynamic simulation of a crystal rod, the solution value of Equation (7) could be used as a reference for the selection of the convective heat transfer coefficient.

## 4. Conclusions

In this paper, a fluid–structure interaction model including a glass tube, cooling channel, coolant and crystal rod is established in ANSYS FLUENT for a configuration with a multiple-LD-array side-pumped laser amplifier. The pumped energy, the glass tube and the crystal rod in this model remain unchanged, but the two initial calculation conditions of the cooling layer thickness and the coolant velocity are changed. The maximum temperature, maximum equivalent stress, inlet pressure and convective heat transfer coefficient are analyzed. The results show that:When the cooling layer thickness changes, the relationship between the maximum temperature (or maximum equivalent stress) and the coolant velocity presents a similar rule. The maximum temperature (or maximum equivalent stress) decreases with the increase in the coolant velocity.When the coolant velocity changes, the larger cooling layer thickness with more coolant at low velocity is not conductive enough to improve heat dissipation for the crystal rod. In the case of high velocity, with df = 1.5 mm as the dividing line, when the cooling layer thickness is above or below 1.5 mm, the influence of cooling layer thickness on the maximum temperature can be ignored.At high velocity, the effect of the cooling layer thickness on the maximum equivalent stress is not very significant. Although the smaller cooling layer thickness will result in a lower maximum temperature and thermal stress, it will significantly increase the inlet pressure. Therefore, the impact of the above factors needs to be weighed comprehensively.The convective heat transfer coefficient results between the simulation and the equation solution are similar, which will guide the solid thermodynamic simulation of only the crystal rod.

The above work provides important references for the design of an LD side-pumped laser amplifier configuration, especially for the thermal management process.

## Figures and Tables

**Figure 1 micromachines-14-00299-f001:**
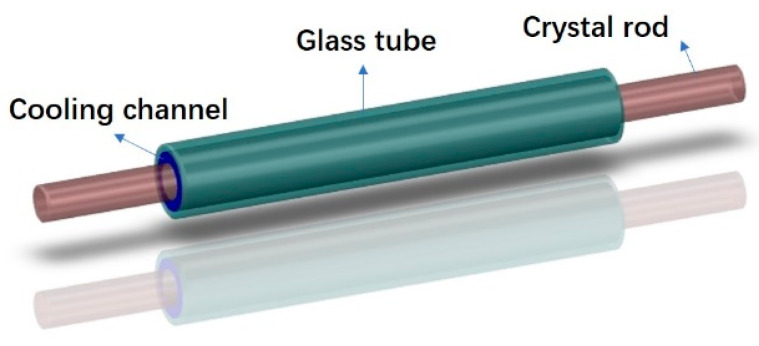
The fluid–structure interaction model.

**Figure 2 micromachines-14-00299-f002:**
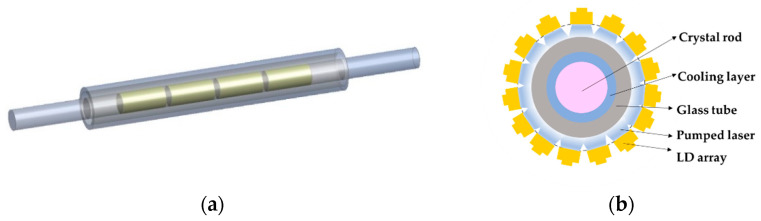
(**a**) LD-array-irradiated regions on the crystal rod (shown in yellow); (**b**) structure with one LD array.

**Figure 3 micromachines-14-00299-f003:**
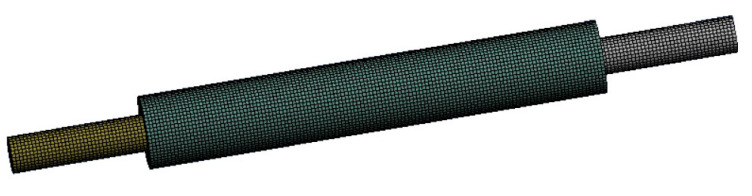
The meshed model for fluid–structure interaction.

**Figure 4 micromachines-14-00299-f004:**
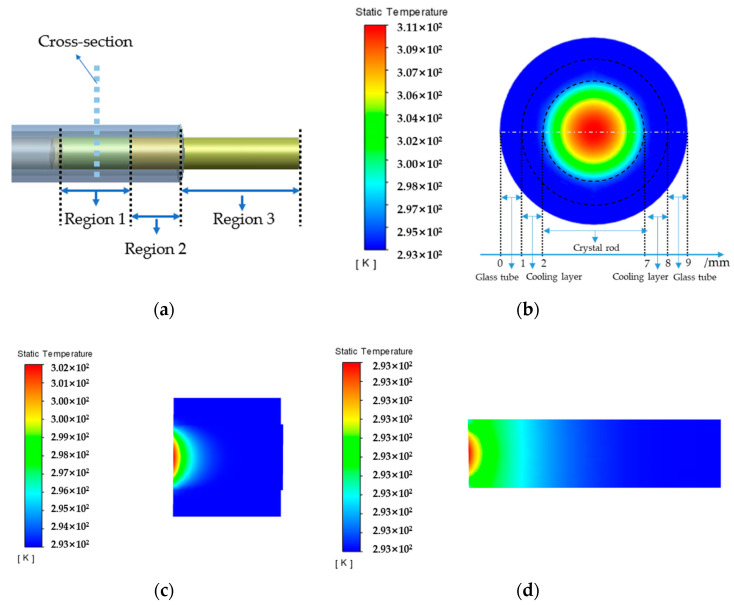
(**a**) The location of the selected cross-section; (**b**) the temperature field in the selected cross-section; (**c**) the temperature profile of Region 2 in the section along the axis; (**d**) the temperature profile of Region 3 in the section along the axis (V = 4.5 m/s, df = 1 mm).

**Figure 5 micromachines-14-00299-f005:**
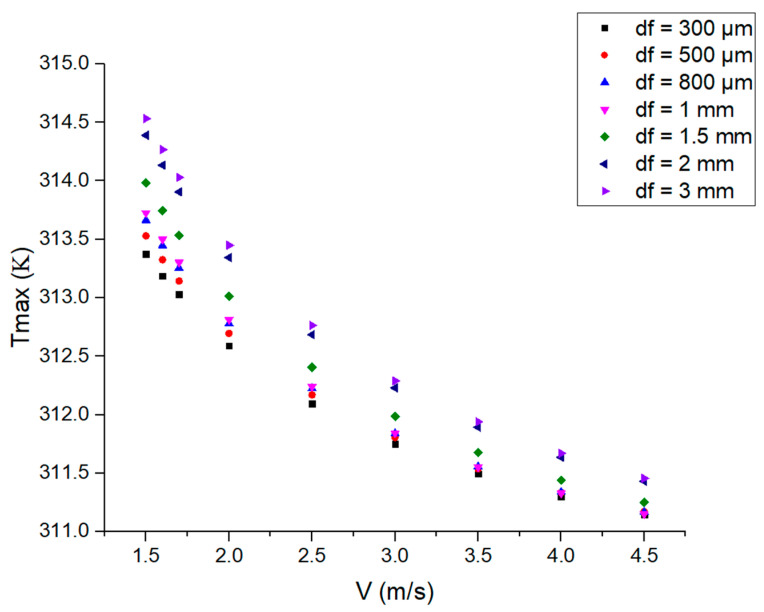
Maximum temperature vs. velocity for various cooling layer thickness values.

**Figure 6 micromachines-14-00299-f006:**
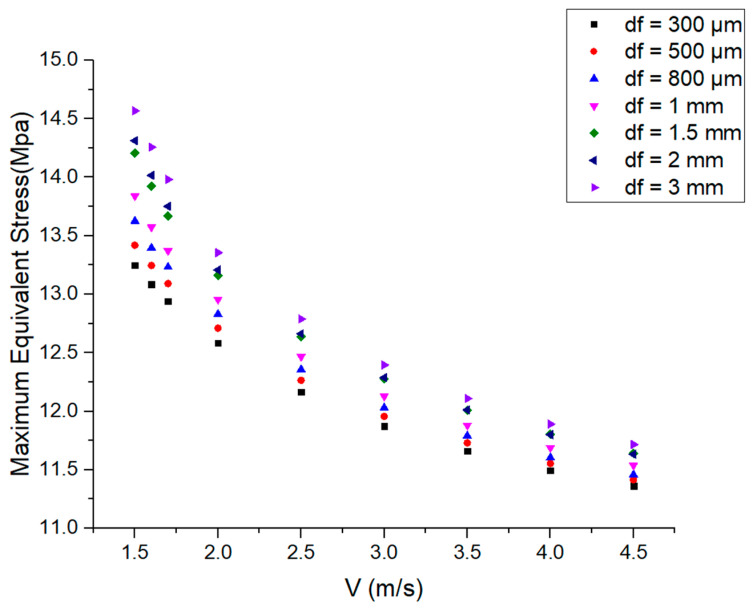
Maximum equivalent stress as a function of coolant velocity for various cooling layer thicknesses.

**Figure 7 micromachines-14-00299-f007:**
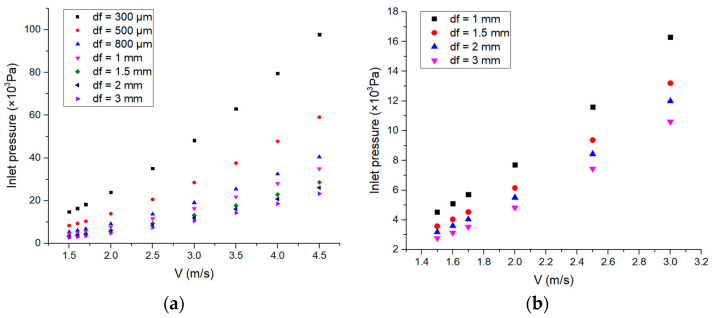
Inlet pressure vs. coolant velocity under different cooling layer thicknesses. (**a**) From 1.5 m/s to 4.5 m/s; (**b**) a range of 1.5 m/s to 3 m/s was considered.

**Figure 8 micromachines-14-00299-f008:**
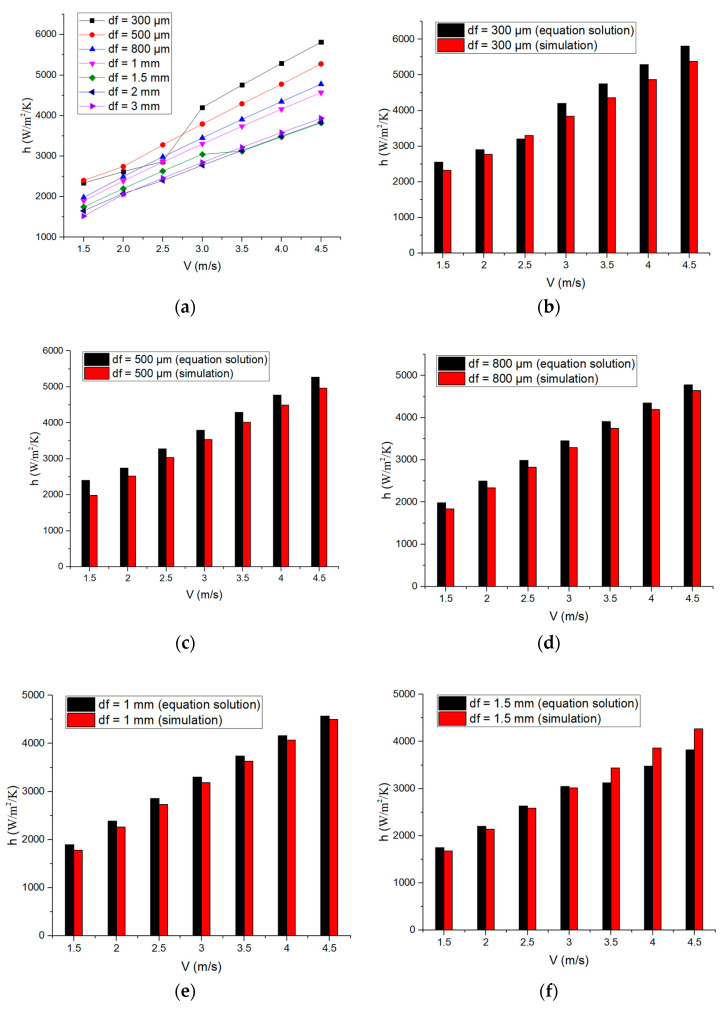
Convective heat transfer coefficient results. (**a**) Equation solution; (**b**) the comparison results for df = 300 μm; (**c**) the comparison results for df = 500 μm; (**d**) the comparison results for df = 800 μm; (**e**) the comparison results for df = 1 mm; (**f**) the comparison results for df = 1.5 mm; (**g**) the comparison results for df = 2 mm; (**h**) the comparison results for df = 3 mm.

**Table 1 micromachines-14-00299-t001:** Material parameters used in the model.

Property	Units	Value
Density (crystal)	kg/m^3^	2830
Thermal conductivity (crystal)	W/m/K	0.558
Specific Heat (crystal)	J/kg/K	750
Coefficient of thermal expansion (crystal)	1/K	1.07 × 10^−5^
Young’s modulus (crystal)	Pa	5.1 × 10^10^
Poisson’s ratio (crystal)	1	0.232
Density (coolant)	kg/m^3^	1793
Thermal conductivity (coolant)	W/m/K	0.063
Specific Heat (coolant)	J/kg/K	1038
Viscosity (coolant)	Kg/m/s	1.359 × 10^−3^
Coefficient of thermal expansion (coolant)	1/K	1.48 × 10^−3^
Density (glass tube)	kg/m^3^	2200
Thermal conductivity (glass tube)	W/m/K	1.4
Specific Heat (glass tube)	J/kg/K	670

## Data Availability

Data are contained within the article.

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
