# Peer review of "The Effect of Cooling Layer Thickness and Coolant Velocity on Crystal Thermodynamic Properties in a Laser Amplifier"

_micromachines, 2023, doi:10.3390/mi14020299_

Round 1
Reviewer 1 Report (New Reviewer)
The manuscript described and further analysed the a fluid-structure interaction model including glass tube, cooling channel, coolant and crystal rod is established in ANSYS FLUENT for the configuration of a LD side pumped laser amplifier. The simulation called into less applying potential. And the meaning of material parameters used in the model were not given. The references cited in this journal seemed to be old, and the introduction should given the key scientific question in LD areas. Therefore, the manuscript is not suitable for publication in this journal.
Author Response
Please see the attachment.

Reviewer 2 Report (Previous Reviewer 3)
The suggested correction given by the reviewers have been correctly addressed. I recommend publication.
Author Response
Thanks for reviewer’s comments and we really appreciate it.
Reviewer 3 Report (New Reviewer)
1.Parameter symbols on line 108 and 109 need to be realigned.
2. In line 136-137, those four groups of LD arrays edmit the same wavelength light or not, Figure 2 should indicate the color and irradiation direction of the corresponding wavelength light.
3. In line 155-157, how to calculate the parameters of cooling channel and coolant velocity, Are these two parameters related?
The whole paper seems to just describe a simulation calculation work of engineering fluid dynamics, and the formulas used are relatively common formulas. If adapted input conditions. Boundary conditions, whether different conclusions will be obtained, whether the results of these simulation calculations will help the optimization and redesign of the system or some kind of innovation, there is no substantive statement.
Round 2
Reviewer 3 Report (New Reviewer)
1. In line 186-188, the number of cooling layer thickness is 7, and the number of coolant velocity is 9, Do these two values correspond
2. In line 201 and 204, the effective numbers of temperature are too long
3. The conclusion part is best to explain that changing the initial conditions and boundary conditions, what are the differences between the finite element simulation results?
Author Response
Please see the attachment.

This manuscript is a resubmission of an earlier submission. The following is a list of the peer review reports and author responses from that submission.
Round 1
Reviewer 1 Report
Since the manuscript is mostly an incoherent text, it is difficult to assess its significance for specialists in this field! In the form in which this manuscript is presented, it is difficult to understand what new the authors are trying to communicate to the scientific community working in this field! For example, the introduction is a set of incoherent pieces of text from which nothing can be understood! Chapter 2.1 is devoted to the description of the theory, which, as its authors understand, should be the basis of the article. Equation (1) is unclear what it describes, not to mention the preservation of dimensionality. What follows is a mention about the Finite Element analysis.
I will not go further into the enumeration of the absurdities given in the reviewed article, but I just want to point out that any further consideration of this text is possible after its coordinated processing!

Reviewer 2 Report
In this paper, the authors establish a fluid-structure coupling thermal analysis model including glass tube, cooling channel, coolant and crystal rod for the configuration of a LD side pumped laser amplifier. This work provides important references for the design of an amplifier configuration, especially for the thermal management process. In my opinion, this work can be accepted for publication after answering the following questions.
11) What’s the possible pump laser wavelength range? In this simulation, one assumption is that the used Coolant and glass tube are transparent (low/no absorption) in this wavelength range – no other thermal effects induced by side pump, right? If yes, such claim should be included in the text.
22) In section 3.2, the simulation results show that the larger the coolant layer thickness, the higher the maximum temperature of the crystal rod. Also, “It can be noted that under the condition of constant low velocity, larger cooling layer thickness with more coolant is not conductive to better dissipate heat for the crystal rod”. What is the possible reason for those phenomenon?
33) It is important that the readers can repeat the calculation/simulation work after reading the paper and in this work some necessary description is missing. E.g. in Section 3.4, the inlet pressure has been calculated. Is it possible to provide the formula or explain briefly how this parameter is calculated?
44) In Section 3.5, it mentioned that ‘V, μ, C, ρ, r is velocity, viscosity, specific Heat, density and coefficient of thermal expansion of the crystal rod, respectively.’ My question: Are all those parameters are of the crystal rod? I thought that the first several parameters should be for coolant, right?
Reviewer 3 Report
The contents of this article is not new and original but nevertheless it is an interesting contribution which may be published once the authors take care of the following points:
1.- In line 27 it is mentioned that DPSSL has several advantages among which it is mentioned "good beam quality" but it should be mentioned that this strongly depends on the spatial and geometric structure of the pump and active media. See for example i) "High repetition rate Q-switching of a high power Nd:YVO4 slab laser" Optics Communications 218 (2003) 155 - 160, and ii) "Experimental study and modelling of a diode-side-pumped Nd:YVO4 laser" Optics Communications 201 (2002) (4-6): 425. In those references you can see the strong asymetries on the beam profile which are produced.
- 2.- You should point out that the active media may be a rod or a slab (rectangular prism)
3.- In line 104 From the convective heat transfer between 1 and 10 you choose 6.5. Why? Based on what ground? On the other hand, Is the irradiation radially symmetric? How do you achieve this?
4,.- Figure 4 is confusing. I suggest to add on the right figure and below it, an axis showing the beginning and end of the cooling channel as well as the thickness of the tube and also the internal rod. The reader should understand that each color corresponds to a different physical position.
5.- What happen in the regions of the rod where there is not coolant? Why don't you show similar graphical results as those of figure 4?
6.- I believe that your study should also include a careful longitudinal analysis. In particular you should explain what happens in the border where the rod has in one side coolant and in the other side air. Could the different temperatures there cause the fracture or even the breaking of the rod?
7.- Even though it is of the outmost importance, you say nothing about the thermal effect of the temperature gradients on the beam profiles. As we know, the heating of the rod near the axis will act as a lens. How important will this be?
Round 2
Reviewer 1 Report
Before evaluating the presented article, the authors should completely (without strikethrough) rewrite both the Abstract and Introduction Sections, as well as give a physical justification for the chosen model, and not refer to the mysterious “Based on Computational Fluid Dynamics and Numerical Heat Transfer”!!!
The physical model that forms the basis of the proposed study should be described! Without this, equation 1 is devoid of any meaning, and the proposed article is only some kind of computational variant. Thus, the physical model must be carefully described with all the resulting limitations. First of all, in the Section 3.1 should carefully describe the calculation procedure, but not the size of the crystal rod and glass tube element!
All equations must end with a comma or a dot!!!
Sec.3.4. From which it follows that “Inlet pressure of the configuration must be considered because high inlet pressure will lead to the sealing and assembly difficulties.”!
Section 3.5 should provide a physical explanation of the choice of the equation of the coefficient of surface convective heat transfer! Any regimes such as “When the Reynolds number is greater than 2100, we replace the diameter inside tubes with the equivalent diameter of annular to approximate the solution in the equation (2)”
must be justified!
The article should not be published in its present form.

Reviewer 3 Report
I do believe that the references suggestions given to the authors (or ANY other with similar results) should be included
Round 3
Reviewer 1 Report
Since the authors stubbornly refuse to follow my recommendations regarding this manuscript (see my previous report), I recommend rejecting this manuscript.
